# Immunomodulatory and pro-oncologic effects of ketamine and isoflurane anesthetics in a murine model

Dominique Abrahams[1☯], Arig Ibrahim-Hashim[1,2☯], Robert S. Ackerman[3,4], Joel S. Brown[2,3,5,6], Christopher J. Whelan[1,6], Megan B. Garfinkel[6], Robert A. Gatenby[7,8], Aaron R. Muncey[3,4]*

1 Department of Cancer Physiology, H. Lee Moffitt Cancer Center and Research Institute, Tampa, Florida, United States of America, 2 Department of Integrative Biology, College of Arts and Sciences, University of South Florida, Tampa, Florida, United States of America, 3 Morsani College of Medicine, University of South Florida, Tampa, Florida, United States of America, 4 Department of Anesthesiology, H. Lee Moffitt Cancer Center and Research Institute, Tampa, Florida, United States of America, 5 Department of Integrated Mathematical Oncology, H. Lee Moffitt Cancer Center and Research Institute, Tampa, Florida, United States of America, 6 Department of Biological Sciences, University of Illinois at Chicago, Chicago, Illinois, United States of America, 7 Department of Radiology, H. Lee Moffitt Cancer Center and Research Institute, Tampa, Florida, United States of America, 8 Department of Cancer Biology and Evolution, H. Lee Moffitt Cancer Center and Research Institute, Tampa, Florida, United States of America

☯ These authors contributed equally to this work.
* Aaron.Muncey@moffitt.org

**Data Availability Statement:** All relevant data are within the paper and its Supporting Information files.

## Abstract

### Introduction

Volatile and intravenous anesthetics may worsen oncologic outcomes in basic science animal models. These effects may be related to suppressed innate and adaptive immunity, decreased immunosurveillance, and disrupted cellular signaling. We hypothesized that anesthetics would promote lung tumor growth via altered immune function in a murine model and tested this using an immunological control group of immunodeficient mice.

### Methods

Lewis lung carcinoma cells were injected via tail vein into C57BL/6 immunocompetent and NSG immunodeficient mice during exposure to isoflurane and ketamine versus controls without anesthesia. Mice were imaged on days 0, 3, 10, and 14 post-tumor cell injection. On day 14, mice were euthanized and organs fixed for metastasis quantification and immunohistochemistry staining. We compared growth of tumors measured from bioluminescent imaging and tumor metastasis in *ex vivo* bioluminescent imaging of lung and liver.

### Results

Metastases were significantly greater for immunocompromised NSG mice than immunocompetent C57BL/6 mice over the 14-day experiment (partial $\eta^2$ = 0.67, 95% CI = 0.54, 0.76). Among immunocompetent mice, metastases were greatest for mice receiving ketamine, intermediate for those receiving isoflurane, and least for control mice (partial $\eta^2$ =

**Funding:** R.A.G. - National Institutes of Health-National Cancer Institute grant 3U54CA193489-05S5, "Cancer as a Complex Adaptive System". https://grantome.com/grant/NIH/U54-CA193489-05S5 The funders had no role in study design, data collection and analysis, decision to publish, or preparation of the manuscript

**Competing interests:** The authors have declared that no competing interests exist.

0.88, 95% CI = 0.82, 0.91). In immunocompetent mice, significantly decreased T lymphocyte (partial $\eta^2$ = 0.83, 95% CI = 0.29, 0.93) and monocyte (partial $\eta^2$ = 0.90, 95% CI = 0.52, 0.96) infiltration was observed in anesthetic-treated mice versus controls.

## Conclusions

The immune system appears central to the pro-metastatic effects of isoflurane and ketamine in a murine model, with decreased T lymphocytes and monocytes likely playing a role.

## Introduction

In the United States, 61.4 percent of patients admitted to the hospital with a cancer diagnosis require a surgical procedure [1]. Despite the fact that most cancer patients will be exposed to anesthetic agents during or even throughout their course of care, the scientific community has only recently gained an understanding that certain anesthetics may have deleterious consequences for cancer progression.

Inhaled anesthetics may impact innate and adaptive immunity [2–4]. Volatile anesthetic agents, such as isoflurane, are associated with decreased natural killer (NK) cell count and cytotoxicity, increased apoptosis of T lymphocytes and decreased lymphocyte function [2–9]. Isoflurane is associated with increased gene expression of vascular endothelial growth factor (VEGF), angiopoietin-1, and interleukin-8; and decreased T helper type 1 to T helper type 2 (Th1/Th2) ratio, suggesting disruption of cancer immunosurveillance [6, 9, 10]. The alteration of innate and adaptive immunity, and proangiogenic effects of anesthetics may accelerate cancer progression.

Ketamine may have negative oncologic effects [11] via reduced NK cell activity, increased helper T cell count [12], lymphocyte apoptosis, and failed dendritic cell maturation [2]. Ketamine can increase regulatory T cell (Treg) expression and the CD4+/CD8+ T lymphocyte ratio. Both inhibit anti-tumor immunity [13, 14]. These studies and current knowledge of immunoediting suggest potential links between anesthetic-induced immunomodulation and worsened cancer outcomes such as increased recurrence, metastasis, and tumor burden [15, 16].

Evidence of immunomodulation as a key link between anesthetic administration and increased metastatic potential must be assessed cautiously because volatile anesthetics possess other potential pro-tumor effects, including upregulation of hypoxia-inducible factors (HIFs) and VEGF, important mediators of angiogenesis, which are key for colonization of target organs [15, 17]. Furthermore, volatile anesthetic agents may increase the concentration of matrix metalloproteinases (MMPs) which are critical for local invasion and extravasation [18–20]. Ketamine, less extensively studied, may possess undiscovered mechanisms that promote metastases.

Evaluating unintended immune and pro-oncologic effects of anesthetic agents poses challenges. *In vitro* research is difficult to translate to clinical situations. Patient studies can have confounding effects of cancer staging, patient conditioning, comorbidities, and varying combinations of anesthetics, sedatives, and analgesics. The pro-metastatic effects of ketamine and isoflurane via immunosuppression have been implicated but not established causally. For instance, based on a large, randomized controlled trial of breast cancer patients having potentially curative primary breast cancer resections from multiple countries, Sessler et al found no reduction in recurrence when a local anesthetic combined with the intravenous anesthetic propofol was used in comparison with a general, volatile anesthetic [21]. For these reasons,

preclinical animal studies will be critical to conducting controlled experiments investigating the role of anesthetics on tumor growth and metastasis [22]. In this study, we tested three hypotheses regarding anesthetic-induced changes in immune function using two mouse strains that vary in their immunocompetence to highlight the potential role of immunomodulation on tumor growth in response to anesthetic administration:

1. Isoflurane and ketamine cause an increase in tumor growth in immunocompetent C57BL/6 mice injected with Lewis lung carcinoma cells compared with controls receiving no anesthetic agent.

2. The increase in tumor burden induced by anesthesia will be more pronounced in immunocompetent mice than in immunocompromised NSG mice, even as the former will show less overall tumor burden than the latter.

3. Increased tumor growth results from inhibitory effects of isoflurane and ketamine on immune function.

## Materials and methods

### Animal models and care

All procedures were approved, and all animals maintained under the University of South Florida Research Integrity & Compliance Institutional Animal Care and Use Committee (IACUC; PROTOCOL #: R IS00004306). The C57BL/6 and NSG mouse strains came from in-house breeding colonies based on breeding pairs acquired from Jackson Laboratory.

Immunocompromised NSG mice have combined immunodeficiencies. They are B and T lymphocyte deficient and functionally deficient in natural killer cells [23]. They provide ideal models for studying tumor biology absent immune function [24]. Immunocompetent C57BL/ 6 mice breed well, have long lifespans, and are a widely used general-purpose strain, providing an appropriate model for comparative studies of immune function [25]. For each mouse strain we used females of 8 weeks age.

We used a syngeneic Lewis lung carcinoma cell line (LL/2-Luc-M38) acquired from Xenogen Corporation and transfected with the plasmid CMV-luc; SV 40-neo in PCI-luc plasmid, whose promotor is CMV, and reporter is luciferase (Caliper Life Sciences). It has high tumorigenicity and compatibility with the innate murine immune system. Both immune and tumor responses can be quantified [26].

### Procedures to ensure minimal animal discomfort

All procedures described in this manuscript have been designed to minimize animal distress/ discomfort. Mice show signs of disseminated disease such as pulmonary metastasis indicated by difficult labored breathing. Mice were observed for specific clinical signs of discomfort (e.g., failure to groom, inactivity, failure to respond to stimuli, isolation from cage mates, shivering, ataxia, shallow, rapid and/or labored breathing, pale mucous membranes, cyanosis, soiled anogenital area, vocalization, head tilt, circling, lack of inquisitiveness, and/or a hunched posture). Specific criteria for humane endpoints for euthanasia were if animals demonstrated any of the signs of distress as described above or lost more than 20% of body weight, in which case they were humanely euthanized immediately. Otherwise, they were humanely euthanized at the experimental endpoint of 14 days. Procedures were performed on animals after induction of isoflurane gas anesthesia to minimize pain/discomfort.

## Methods of euthanasia

The method used for euthanasia is consistent with the recommendation of the Panel on Euthanasia of the American Veterinary Medical Association. Mice were euthanized by inhalation of lethal doses of CO2 by gradually increasing concentrations of CO2 (e.g., 33% achieved after 1 minute) from a compressed gas source. Death was verified by employing a cervical dislocation.

## Anesthetics and reagents

We administered inhaled isoflurane (Henry Schein, Melville, NY) mixed with oxygen at a 2–3% flow rate. We administered ketamine (80mg/kg) mixed with 10 mg/kg Xylazine (Covetrus, Portland, Me) via an intraperitoneal (IP) injection. D-Luciferin, sodium salt (Gold Biotech, St. Louis, MO) was prepared with sterile phosphate buffered saline at 15mg/ml for bioluminescent imaging.

## Experimental and control groups

We employed a $2 \times 3$ factorial design which crossed mouse strain (C57BL/6 and NSG mice) by anesthetic treatment. A total of 15 mice of each strain were randomly subdivided into three groups (isoflurane, ketamine, control), yielding 5 mice per each combination of mouse strain × anesthetic (Fig 1). Because the LL/2-Luc-M38 is well characterized by Caliper Life science, with a metastasis rate of 100%, we used the same sample size (n = 5) that we have used in previous publications, in which tumor cells were inoculated into mice (intravenously and into the mammary fat pad) and the response to treatment was measured with bioluminescence [27–29]. The whole experiment was repeated three times. To reduce bias, the quantifications of positive pixels were performed by a person who was blind to treatment allocations.

## Cell culture and inoculation

The LL/2-Luc-M38 cells were maintained at 5% $CO_2$ and grown in DMEM/F12 supplemented with 10% fetal bovine serum (HyClone) and 1% pen strep (Corning). For cell injections, $5 \times 10^5$ cells in 200µL PBS were slowly injected intravenously with a 27 Gauge needle. The ketamine group received their cancer inoculation 10 minutes after receiving IP ketamine (80mg/kg)/ Xylazine (10mg/kg). The isoflurane group received their cancer inoculation after anesthesia with isoflurane for 10 minutes at 3% flow rate. The control mice simply received their inoculation of cancer cells. The ketamine and isoflurane groups stayed under their respective anesthesia for an additional 20 minutes post injection of cancer cells. All mice were imaged (IVIS-200) to confirm successful intravenous injections of cancer cells after intra-peritoneal injections of luciferase.

Mice were maintained for 14 days under standard laboratory conditions including *ad libitum* food and water. During that period, they underwent fluorescent imaging at defined time points.

## Tumor growth and metastasis

**Bioluminescent imaging over time.**   We used *in vivo* bioluminescence imaging to quantify tumor growth using the Xenogen IVIS-200 System (Perkin Elmer, Waltham, MA). Prior to each imaging session, mice were IP injected with sterile d-luciferin at 10µl per gram body weight. D-luciferin was prepared in PBS at 15mg/ml. After mice were injected, they were placed inside an oxygen rich induction chamber consisting of 2.5–3% isoflurane. All mice, regardless of anesthesia treatment, received the isoflurane as a necessity for performing imaging. Any effects of these repeated exposures would indiscernibly manifest across all treatment

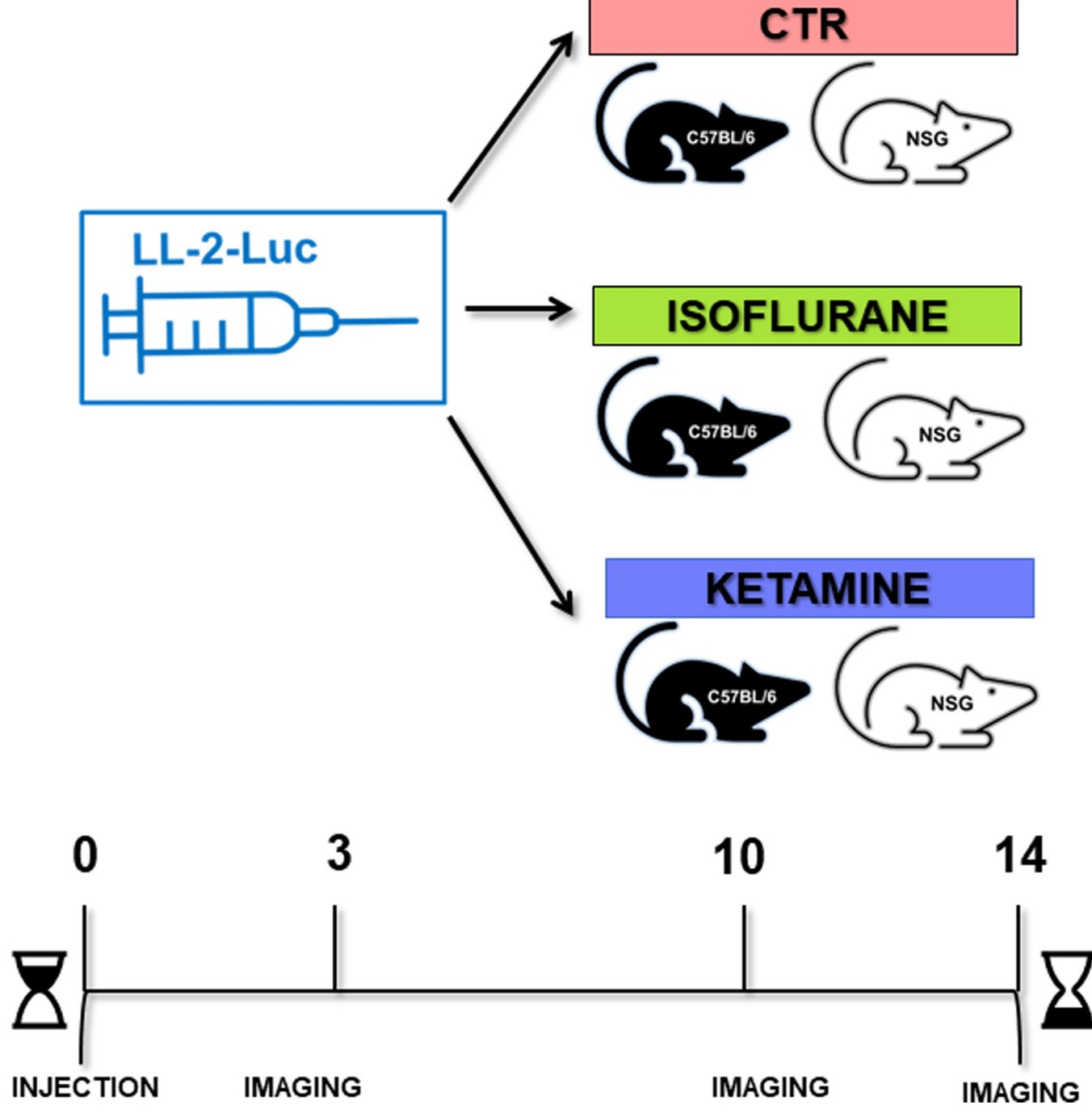

**Fig 1. Study design.** Lewis lung carcinoma (LL-2-Luc-M38) cells were injected via the tail vein into immunocompetent C57BL/6 and immunodeficient NSG mice. Within each population, one control group was kept without anesthesia administration (control). One group was administered volatile anesthesia with isoflurane, and one group was administered intraperitoneal ketamine. Using bioluminescence methods, the mice were imaged on days 0, 3, 10, and 14 post tumor cells injection.

groups. The mice were imaged for 5 minutes post injection of d-luciferin. Mice were placed on their backs with ventral side up on the imaging platform. Anesthesia was maintained using nose cones with a 2% isoflurane flow rate. The IVIS imaging chamber consisted of a warming

platform and a cryogenically cooled CCD camera to capture both a visible light photograph and a bioluminescent image. Imaging time points included at the time of injection (time 0) and on days 3, 10, and 14 post injection.

***Ex vivo* bioluminescence.** At day 14, mice were euthanized followed by *ex vivo* imaging of the lungs and liver. Data were acquired and analyzed utilizing Living Imaging 4.7.4 software. Photon intensity (p/s) was assessed in regions of interest in the chest area (*in vivo*) and lungs (*ex vivo*).

**Histology and immunohistochemistry (IHC).** Following *ex vivo* bioluminescence imaging of euthanized mice, lungs, liver, and kidneys were harvested and fixed in 10% neutral buffered formalin. Tissues were then processed, embedded in paraffin, and sliced into 4 - 5 μm sections. Slides from lung and liver were stained with hematoxylin and eosin (H&E) and graded by a pathologist for the presence of metastatic tissue. We also performed immunohistochemistry for CD3 and CD11b antibodies on a consecutive slide from lungs.

**Immunohistochemistry for CD3 and CD11b antibodies.** Slides from the lung sections were deparaffinized with EZ Prep solution (Ventana). The heat-induced antigen retrieval method was used in Cell Conditioning 1 Mild (Ventana). The primary rabbit antibody that reacts to CD3 (ab16669, Abcam, Cambridge, MA) was used at a 1:200 dilution in Dako antibody diluent (Carpenteria, CA) and incubated for 32 minutes. The primary rabbit antibody that reacts to CD11b (#LS-C141892, Lifespan Bioscience, Seattle, WA) was used at a 1:700 dilution in Dako antibody diluent (Carpenteria, CA) and incubated for 28 minutes. For both antibodies, the tissue section was exposed to Ventana OmniMap Anti-Rabbit Secondary Antibody for 16 minutes. The detection system used was the Ventana ChromoMap kit, and slides were then counterstained with Hematoxylin. Slides were dehydrated and placed under cover slips as per standard laboratory protocol.

**Image analysis of metastasis.** We used Visiopharm software version 2020 to measure and quantify lung and liver metastases from the tissue cross sections of all mice. H&E images were imported into the software and a threshold segmentation was applied to the hematoxylin channel (with 13x13 median filter) to distinguish tumor from non-tumor areas. The automated segmentation results were corrected by an experienced image-analyst using the software's manual annotation tools. For each sample, the area of tumor and tissue were extracted from the analysis and used to determine the percentage of tissue area comprised of metastases.

**IHC quantification.** IHC slides stained for CD3 and CD11b were scanned using an Aperio AT2 digital pathology system (Leica Biosystems Inc., Vista, California) with a 20X 0.7NA objective lens. CD3 and CD11b positive cells were identified using size and morphology adjustments to Aperio's default Nuclear algorithm. The analysis provided cell counts for each biomarker segmenting results into four categories of staining intensity (0, 1+, 2+, 3+) according the Aperio's default thresholds for scoring. Using the percentages of positive cells for each category, H-Scores were determined for each sample using the published formula: H-score = (% of cells stained at intensity category 1) + (% of cells intensity category 2 × 2) + (% of cells intensity category 3 × 3) [30].

## Statistical analysis

To determine if tumor growth over the 14-day experiment varied by mouse strain and anesthetic, we conducted a repeated-measures analysis of variance (rmANOVA) for combined data from the three repeats of the experiments (Table 1), using total photon flux values for days 3, 10 and 14 [31, 32]. We used a natural log transformation of the total flux to better meet the assumption of normality and to linearize the exponential growth rates of the cancer cells. We constructed a maximal model with ln(total flux) as the dependent variable, and

**Table 1.** Repeated measures analysis of variance examining effects of experiment (Expt), mouse strain (C57BL/6, NSG), and anesthesia treatment (anesthetic: Control, isoflurane, ketamine) on whole organism bioluminescence (photon flux) over the course of experiments.

| Effect | Df | MSE | F | P |
|---|---|---|---|---|
| **Between Subjects** | | | | |
| Expt | 2, 66 | 1.25 | 67.94 | <**0.001** |
| Mouse Strain | 1, 66 | 1.25 | 462.87 | <**0.001** |
| Anesthetic | 2, 66 | 1.25 | 5.69 | **0.005** |
| Expt×Mouse Strain | 2, 66 | 1.25 | 4.43 | **0.016** |
| Expt×Anesthetic | 4, 66 | 1.25 | 4.41 | **0.003** |
| Mouse Strain×Anesthetic | 2, 66 | 1.25 | 1.21 | 0.306 |
| **Within Subjects** | | | | |
| Day | 1.84, 121.33 | 1.03 | 337.21 | <**0.001** |
| Expt×Day | 3.68, 121.33 | 1.03 | 6.84 | <**0.001** |
| Mouse Strain×Day | 1.84, 121.33 | 1.03 | 42.75 | <**0.001** |
| Anesthetic×Day | 3.68, 121.33 | 1.03 | 1.88 | 0.124 |
| Expt×Mouse Strain×Day | 3.68, 121.33 | 1.03 | 7.28 | <**0.001** |
| Expt×Anesthetic×Day | 7.35, 121.33 | 1.03 | 1.41 | 0.203 |
| Mouse Strain×Anesthetic×Day | 3.68, 121.33 | 1.03 | 2.55 | **0.047** |

experiment, anesthetic, and mouse strain as independent variables (main effects), day as the repeated measure, and all two-way and three-way interactions. To run the rmANOVA, we used the package *afex* in program R [33]. Package *afex* automatically applies the Greenhouse-Geisser sphericity correction to factors violating the sphericity assumption.

To determine if the metastasis to liver and lung as measured by bioluminescence of whole organ varied by mouse strain and anesthetic, we conducted an analysis of variance (ANOVA) of the *ex vivo* organ photon flux for the combined data from the three repeats of the experiment (Table 2). We used a natural log transformation of total flux to better meet the assumption of normality and to linearize exponential growth of the cancer cells. We initially constructed a maximal 4-way ANOVA with ln(total flux) as the dependent variable, and experiment, anesthetic, mouse strain, and organ type as independent variables (main effects), and all possible interactions of the main effects. We simplified the models by eliminating non-significant interactions found in the maximal model. The final model included ln(total flux) as the dependent variable, experiment, mouse strain, and organ type as main effects, and the interactions of experiment × mouse strain, experiment × organ and mouse strain × organ type.

To determine the effect of experiment, anesthetic, mouse model, sampled organ (liver or lung), and interactions of these variables on the proportion of tissue sample consisting of

**Table 2.** Analysis of variance on effects of experiment (Expt), mouse strain (C57BL/6, NSG), and organ (liver, lung) type on *ex vivo* bioluminescence (photon flux).

| Effect | df | Sum Sq | Mean Sq | F | P |
|---|---|---|---|---|---|
| Expt | 1 | 68.7 | 68.7 | 7.72 | **0.006** |
| Mouse Strain | 1 | 11140.9 | 11140.9 | 128.21 | <**0.001** |
| Organ | 1 | 509.5 | 509.5 | 57.25 | <**0.001** |
| Expt×Mouse Strain | 1 | 112.5 | 112.5 | 12.64 | **0.005** |
| Expt×Organ | 1 | 72.6 | 72.6 | 8.16 | **0.004** |
| Mouse Strain ×Organ | 1 | 53.4 | 53.4 | 6.00 | **0.015** |
| Residuals | 153 | 1361.6 | 8.9 | | |

metastasis we conducted a zero-inflated beta distribution regression (Table 3). Because the response variable included numerous zero values (zero-inflated), we conducted an analysis that uses a piecewise distribution to model both the probability that there is no metastasis (prY = 0), as well as the effect of each predictor variable on the magnitude of metastasis when y>0. To accomplish this, we used a Bayesian inference for zero-inflated beta regression (zoib) model run with package *zoib* (version 1.5.1) in program R [31, 34]. The model was run with 2 chains, 1,000 iterations burn-in, 20,000 iterations post burn-in, and thinning set to 20. Convergence of the chains was checked by inspection of trace plots and the potential scale reduction factors [35].

To determine effects of anesthesia treatment on CD3 and CD11b staining in C57BL/6 mice, we used a one-way between subjects analysis of variance (ANOVA). A Dunnett's test identified the pairs of treatments that showed significant differences. NSG mice were excluded from this analysis owing to their lack of mature T and B lymphocytes, which preclude positive staining in tissue sections

## Results

### Tumor growth and metastasis

**Bioluminescent imaging.** Based on whole organism bioluminescence, immunocompetent C57BL/6 mice had significantly lower metastatic burden than immunocompromised NSG mice (Fig 2A). In C57BL/6 mice, treatment with ketamine significantly increased the metastatic burden in comparison to the other treatments. For NSG mice, metastatic burden did not vary with anesthesia treatment (Fig 2A) as there was no statistically significant difference in flux amongst the different anesthetic treatments at each of the three time points. *Ex vivo* images from lung, liver, and kidneys from each anesthesia treatment illustrate these results as well as larger metastases to lungs in comparison to liver and kidney (Fig 2B).

The rmANOVA of tumor growth (total flux) over time identified significant effects of experiment, mouse strain, anesthetic, and day (the repeated measure) (Table 1). Tumor growth was significantly greater in mice of the second repeat of the experiment than in the other two. Tumor growth rates were significantly greater in the NSG than in the C57BL/6 mice, and tumor growth rates were significantly greater in C57BL/6 mice administered with ketamine and isoflurane compared to the control.

The significant two- and three-way interactions (Table 1) indicate that the relative differences in tumor growth in NSG mice compared to C57BL/6 mice (Fig 3A) were contingent on time and anesthetic treatment (Fig 3B).

*Ex vivo* **bioluminescence.** The ANOVA model of *ex vivo* metastatic burden (photon flux) provided a good fit to the data (multiple $R^2$ = 0.589). *Ex vivo* metastatic burden showed significant effects of experiment, mouse strain, and organ type, but not anesthesia treatments (Table 2). The significant interactions of experiment × mouse strain, experiment × organ, and mouse strain × organ type (Table 2) indicated that *ex vivo* metastatic burdens were significantly greater in experiment 2 than the other two experiments, greater in NSG than C57BL/6 mice, and greater in lung than in liver. The greater *ex vivo* metastatic burden in NSG than in C57BL/6 mice was more pronounced in experiment 2 than the others (Fig 4A). The greater *ex vivo* metastatic burden in the lung than in liver was less pronounced in NSG than in C57BL/6 mice (Fig 4B).

**Image analysis of metastasis.** The zoib model indicated that the proportion of tissue consisting of tumor differed among experiments, between mouse strains, and between organ types, but did not differ among the anesthetic treatments (Table 3). The beta regression coefficients for which the 95% credible interval do not overlap zero indicate that proportion of

**Table 3. Posterior inferences of the coefficients (on the logit-scale) in the best Bayesian zero-one-inflated beta distribution model on proportion of tissue area with tumor metastasis.** The first model component estimates the mean (linear predictor) in the model, and the second component the probability of zero. The factor levels 'Experiment 1', 'Control' (anesthetic), 'C57BL/6' (mouse strain) and 'Liver' (organ) are the baseline values in the model and are included in the intercepts. d—Regression coefficient in the linear predictor for the sum of the two shape parameters in the beta distribution. *—indicates a significant difference when the quantile range does not overlap zero.

| Model Component | Coefficient | Mean | 25% quantile | 97.5% quantile |
|---|---|---|---|---|
| Logit(mean) | Intercept* | -4.338 | -5.133 | -3.568 |
| | Experiment 2* | 0.918 | 0.531 | 1.246 |
| | Experiment 4 | 0.221 | 0.074 | 0.567 |
| | Isoflurane | 0.030 | -1.066 | 1.172 |
| | Ketamine | 0.527 | 0.584 | 1.685 |
| | NSG* | 1.768 | 0.966 | 2.637 |
| | Lung* | 2.544 | 1.607 | 3.560 |
| | Isoflurane:NSG | 0.284 | -0.639 | 1.067 |
| | Ketamine:NSG | 0.135 | -0.699 | 1.158 |
| | NSG:Lung | 0.494 | 0.366 | 1.293 |
| | Isoflurane:Lung | -0.490 | -1.403 | 0.402 |
| | Ketamine:Lung | -0.809 | -1.698 | 0.230 |
| logit(Pr(y = 0)) | Intercept | -0.419 | -1.113 | 0.0463 |
| | Isoflurane | -0.165 | -1.266 | 0.864 |
| | Ketamine | -0.033 | -1.012 | 1.111 |
| | Lung* | -2.609 | -4.016 | -1.406 |
| | d | 2.131 | 1.880 | 2.454 |

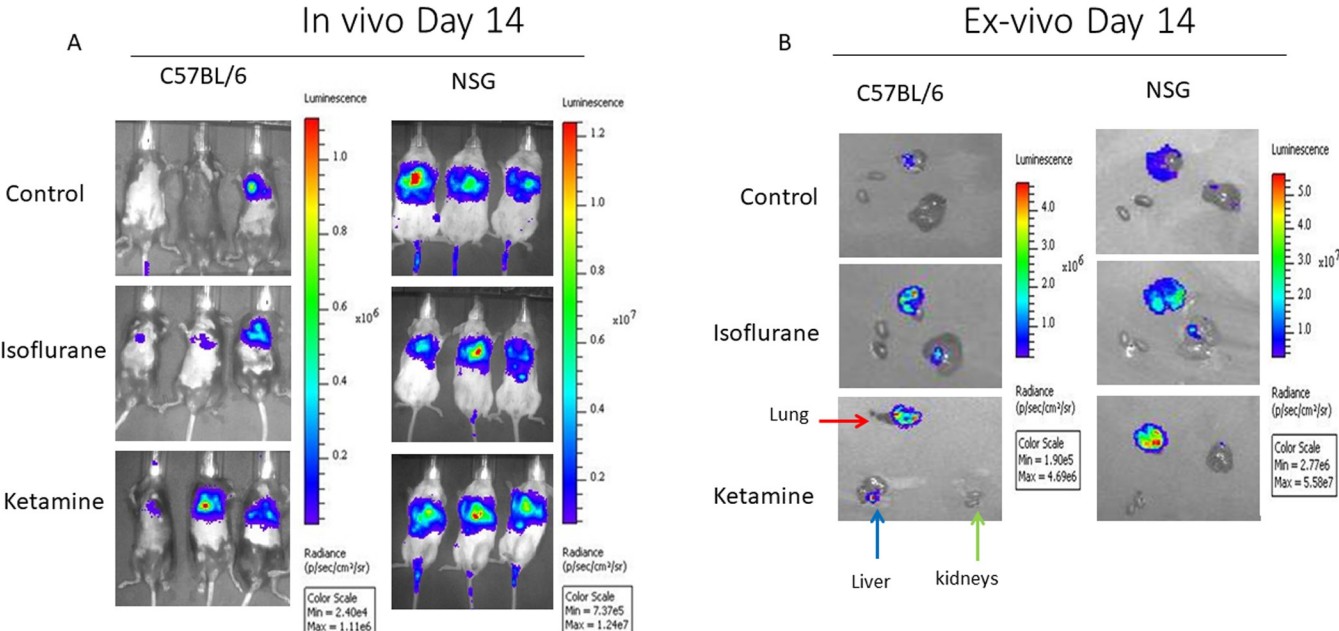

**Fig 2. Bioluminescence images.** A. *In vivo* representative bioluminescence ventral view images of mice from the three treatment cohorts; control, isoflurane, and ketamine in C57BL/6 and NSG mice strains. Images are at day 14 post-injection. Bioluminescence signal is more visible in ketamine treated mice in C57BL/6 strain compared to other treatment groups. Bioluminescence signal is equally visible in all treatment groups in the NSG strain. B. *Ex vivo* representative images of the lung (red arrow), liver (blue arrow), and kidneys (green arrows) from the three treatment cohorts for C57BL/6 and NSG strains. Note the different scale bars in C57BL/6 and NSG.

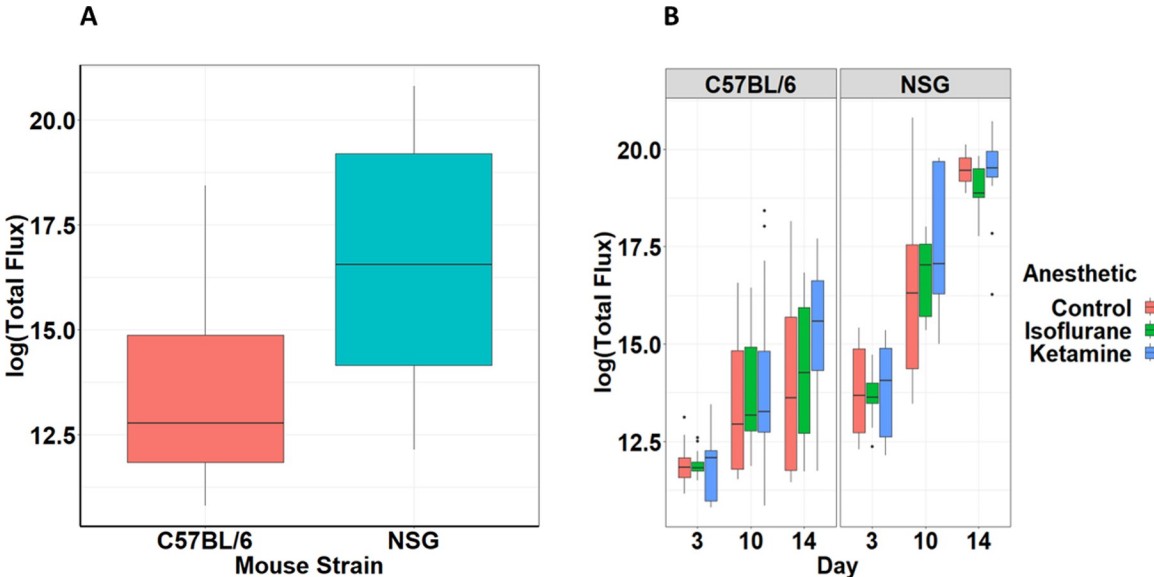

**Fig 3. Quantification of mouse strain and anesthetic effect on metastasis.** A. Natural log of total flux based on bioluminescence imaging of whole organism for C57BL/6 and NSG mouse subjects (5 mice per group, 3 experimental replications). Mean ± SEM. Repeated-measures ANOVA, ($F_{1,66} = 462.87$, $p < 0.001$) B. Natural log of total flux based on bioluminescence imaging of whole organism over the 14-day course of the experiment for C57BL/6 and NSG mouse subjects contingent upon anesthesia treatment. Mean ± SEM. Repeated-measures ANOVA, ($F_{3.68,121.33} = 2.55$, $p = 0.047$).

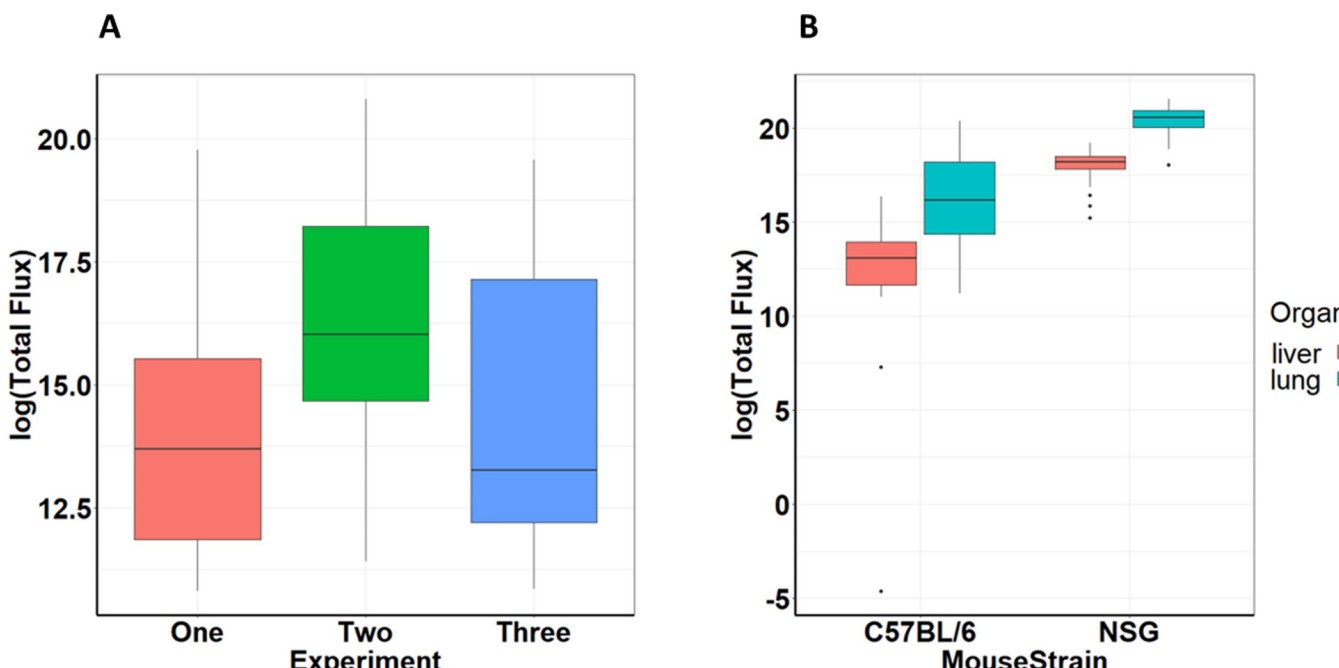

**Fig 4. Quantification of experiment and mouse strain effect on *ex vivo* metastasis.** A. Natural log of total flux based on *ex vivo* bioluminescence imaging for experiments one, two and three (5 mice per group in each experiment). Mean ± SEM. Repeated-measures ANOVA, ($F_{2,66} = 67.94$, $p < 0.001$) B. Natural log of total flux based on *ex vivo* bioluminescence imaging on day 14 of lungs and liver for C57BL/6 and NSG mouse subjects contingent upon mouse strain. Mean ± SEM. One-way ANOVA, ($F_{1,153} = 6.00$, $p < 0.015$).

tumor was greater in Experiment 2 than Experiment 1, was greater in NSG than C57BL/6 mice, and was greater in lung than in liver (Table 3).

The zoib model indicated that anesthetic type had no effect on the absence of cancer in the tissue sample (0% tissue consisting of tumor), but samples from lung were less likely than samples from liver to exhibit no cancer (Table 3).

**Histology and immunohistochemistry.** H&E staining of lungs from C57BL/6 are shown in Fig 5A and from NSG mice in Fig 5B. Because NSG mice lack mature T and B lymphocytes, we did not observe any positive stained cells in the lung sections (Fig 6). Because C57BL/6 mice have an intact immune system, we observed positive cell staining in lung sections. CD3 staining indicated that T cell counts were significantly lower for ketamine and isoflurane treatments than for the control (Fig 7A and 7B). CD11b staining indicated that monocytes were present in NSG and C57BL/6 mice; hence, positive staining was detected on lung sections from both strains in the three treatment groups. No significant changes were observed between anesthesia treatment groups in NSG mice (Fig 8). In C57BL/6 mice the index for monocytes was significantly lower for ketamine and isoflurane treatments than for controls (Fig 9A and 9B).

## Discussion

We found that both ketamine and isoflurane increased overall tumor burden and accelerated tumor growth in immunocompetent, but not immunocompromised mice. These effects emerged even though exposure to either anesthetic was restricted to 30 minutes during tumor cell inoculation compared to control subjects, and outcomes spanned 14 days. That immunocompromised NSG mice demonstrated increased overall tumor burden irrespective of anesthetic exposure was expected given known mechanisms of immune function in cancer surveillance [16]. By day 14, tumor growth in control mice in the immunodeficient cohort did not differ from mice in either anesthetic treatment. By contrast, tumor growth in

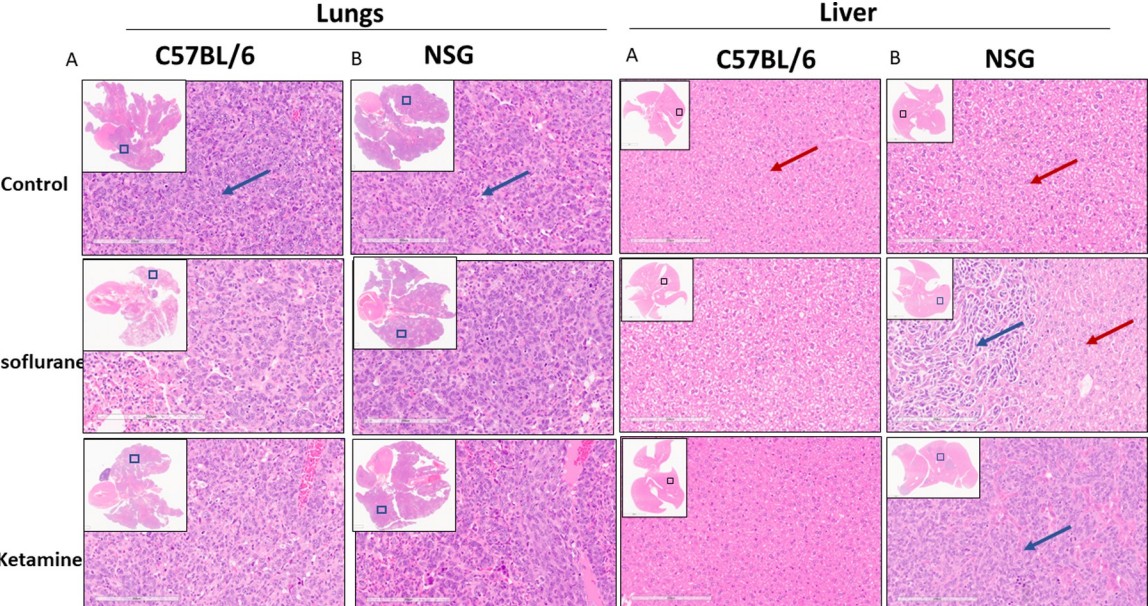

**Fig 5. Representative histologic images (H&E staining) of lung and liver metastasis.** A. Carried by immunocompetent C57BL//6 mice and B. Carried by immune-compromised NSG mice. The representative images are from the three treatment cohorts; the upper panels are treated with no anesthesia (control), the middle panels are treated with isoflurane, and the lower panels are treated with ketamine. Representative high-power fields with inset low-power images of the whole tumor cross-section are shown.

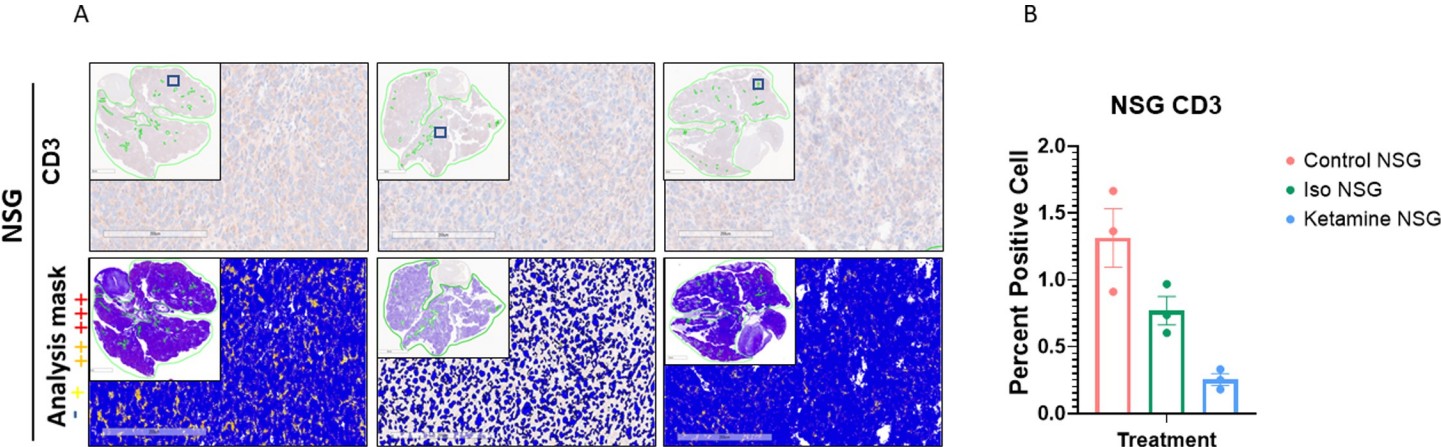

**Fig 6. CD3 staining and quantification.** Representative images of immunohistochemistry staining of CD3 in lungs of A. NSG mice. Images are from the three treatment cohorts (control, isoflurane, and ketamine) and contain representative high-power fields with inset low-power images of the entire tumor cross-section. Positivity mask in lower panels. Percent positive cells quantified over an entire viable area of lung cross-section of B. NSG mice. Mean ± SEM, n = 3 mice each arm. One-way ANOVA, (F2,6 = 14.86, p = 0.0728). p = 0.0796 vs. control group.

immunocompetent mice was dependent upon anesthetic choice and increased with ketamine and isoflurane over time. Taken together, these effects indicate the primary mechanism of anesthetic-mediated increase in tumor growth and tumor burden for both agents was immune modulation.

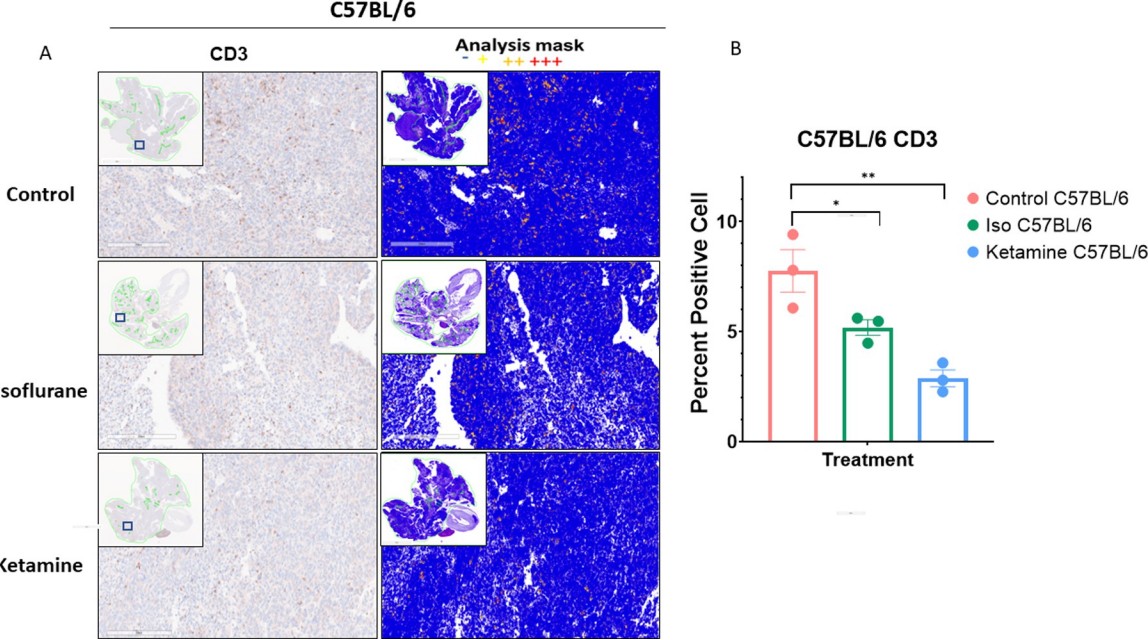

**Fig 7. CD3 staining and quantification.** Representative images of immunohistochemistry staining of CD3 in lungs of A. C57BL/6 mice. Images are from the three treatment cohorts (control, isoflurane, and ketamine) and contain representative high-power fields with inset low-power images of the entire tumor cross-section. Positivity mask in lower panels. Percent positive cells quantified over an entire viable area of lung cross-section of B. C57BL/6. Mean ± SEM, n = 3 mice in each arm. One-way ANOVA, (F2,6 = 14.86, p = 0.0047). **p < 0.0029 and *p < 0.0495 vs. control group.

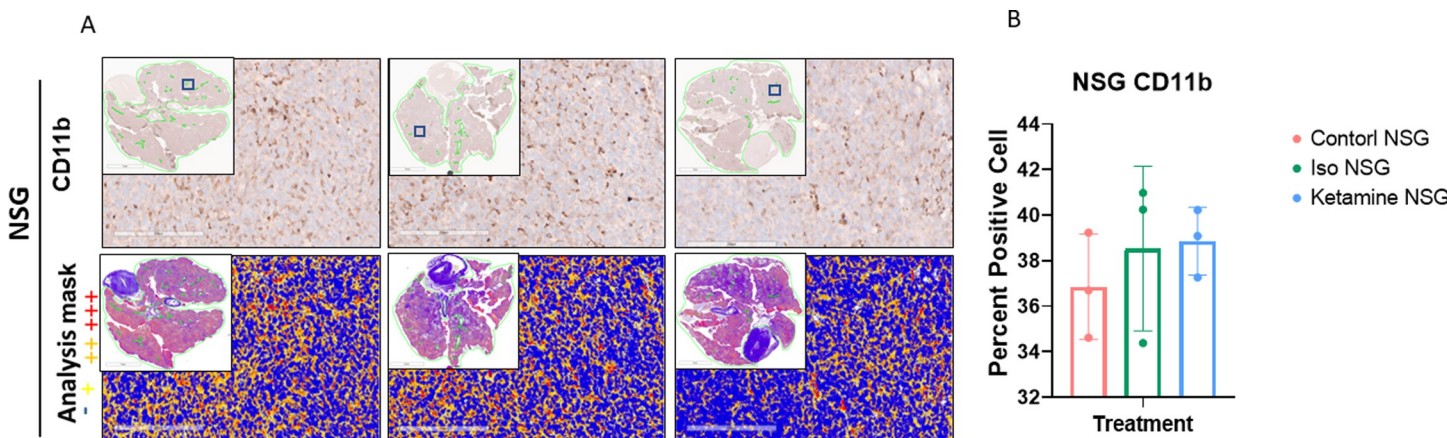

**Fig 8. CD11b staining and quantification.** Representative images of immunohistochemistry staining of CD11b in lungs of A. NSG mice. Images are from the three treatment cohorts (control, isoflurane, and ketamine) and contain representative high-power fields with inset low-power images of the entire tumor cross-section. Positivity mask in lower panels. Percent positive cells quantified over the entire area of lung cross-section of B. NSG mice. Mean ± SEM, n = 3 mice in each arm. One-way ANOVA, ($F_{2,6}$ = 0.5066, p = 0.6262). p = 0.6667 and p = 0.5752 vs. control group.

Our findings corroborate those from a study showing that ketamine and the volatile anesthetic halothane increased metastases, ketamine more so than halothane, in a MADB106 rat model of metastasis by tail vein injection [36]. In this study, animals received 1 hour of

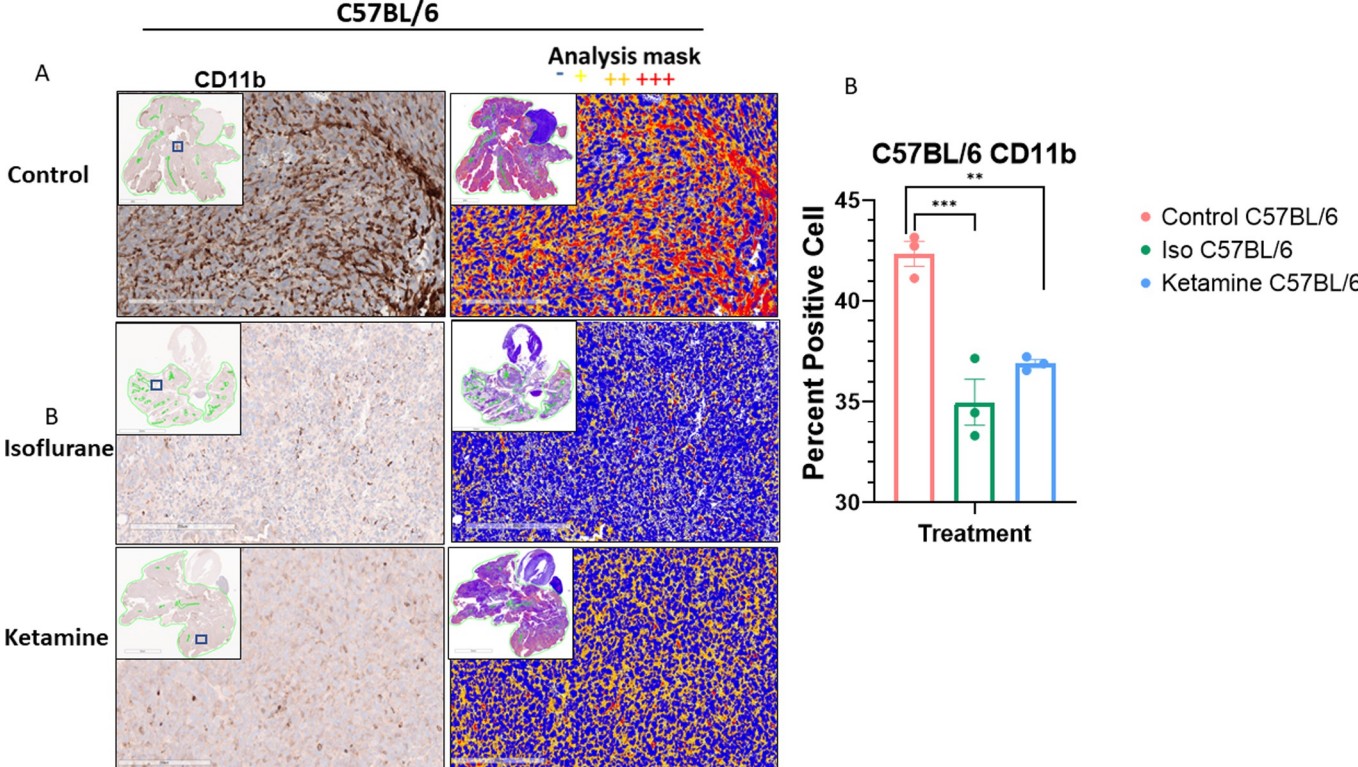

**Fig 9. CD11b staining and quantification.** Representative images of immunohistochemistry staining of CD11b in lungs of A. C57BL/6. Images are from the three treatment cohorts (control, isoflurane, and ketamine) and contain representative high-power fields with inset low-power images of the entire tumor cross-section. Positivity mask in lower panels. Percent positive cells quantified over the entire area of lung cross-section of B. C57BL/6 mice. Mean±SEM, n = 3 mice in each arm. One-way ANOVA, ($F_{2,6}$ = 25.59, p = 0.0012). ***p < 0.0008 and **p < 0.004 vs. control group.

anesthetic exposure, and MADB106 tumor cells were injected 4 hours after anesthetic termination. In our study, tail vein injections occurred during anesthesia (10 minutes into 30 minutes total exposure time) to better ascertain direct and indirect anesthetic effects.

Because we hypothesized that anesthetics increase the likelihood of further tumor growth and/or metastasis mechanistically through their inhibitory effects on immune function, our study included the use of the NSG immunocompromised strain of mice. We found that ketamine and isoflurane-treated mice exhibited increased metastasis in immunocompetent C57BL/6 but not immunocompromised NSG mice, implicating the hypothesized link to immune-inhibition by anesthetic agents.

Histology and immunohistochemistry showed that both anesthetic agents reduced lymphocyte and monocyte infiltration. Based on CD3 staining, immunocompetent mice receiving ketamine and isoflurane had significantly fewer T cells than control mice. Ketamine decreased T cell infiltration more than isoflurane. Isoflurane and ketamine also decreased monocytes in the immunocompetent C57BL/6 mice compared to controls based on CD11b staining, with isoflurane causing a larger decrease than ketamine. There were no significant differences in monocytes in the immunodeficient NSG mice, which lack mature lymphocytes. The decreased infiltration of both T lymphocytes and monocytes in immunocompetent mice receiving either anesthetic agent, coupled with no effect in immunocompromised mice, again implicate an immune inhibitory effect of anesthesia. We conclude that each anesthetic increased the metastatic potential of injected Lewis lung carcinoma via disrupted immune function involving both innate and adaptive immunity.

The metastatic process is inefficient, and previous authors have described how these inefficiencies create temporal patterns in metastasis [37–39]. While our model is relatively simple and does not reflect the entirety of the metastatic cascade, nor the full complexity of the tumor micro-environment or primary tumor effects, we still glean important effects of anesthetic administration on both circulating and disseminated tumor cells which were injected via tail vein. A lung model of melanoma in C57BL/6 mice found that after tail vein injection, multicellular tumor clusters were not detected in lung until 4 days post-injection, with only single disseminated tumor cells detected prior [37]. In our study, tumor cells were injected during anesthesia and a significant difference in metastatic burden between isoflurane or ketamine and controls in C57BL/6 mice was not seen until 10 days post injection. Image analysis revealed that the distribution of metastases did not differ with anesthesia treatment. Thus, the areas that originally seeded with tumor cells were the same areas that developed disease burden. Our findings provide more evidence that angiogenic factors, which develop after tumor cells begin to cluster are not as important as factors affecting immunosurveillance. In Melamed et al., a significant increase in metastasis was found three weeks after MADB106 cells were injected four hours after termination of an hour of ketamine exposure [36]. This is also consistent with a timeframe where immunoediting would be a significant factor in preventing metastasis.

A key strength of our study was the inclusion of an immunological control strain of immunodeficient NSG mice in addition to the immunocompetent C57BL/6 mice to test the hypothesis that anesthetic agents increased tumor growth and burden via immunomodulation. A review of animal studies of the effect of morphine on tumor growth and metastasis [22] reported on 16 studies using either immunocompetent mice or rats, or immunocompromised mice or rats, but not both. Of those 16 studies, 14 employed immunocompetent animal models, and 2 employed immunocompromised animal models. Furthermore, our procedures ensured well-controlled environmental factors, elimination of potential confounding effects, and rigorous statistical analyses.

Nonetheless, weaknesses remain. By using tail vein injections, this and similar studies do not provide an accurate representation of the metastatic cascade as they do not include the stages of epithelial-mesenchymal transition or intravasation. This model is also limited in that it lacks a primary tumor to pre-condition draining lymph nodes, promote T cell activation or suppression, or drive an immune suppressive environment prior to the anesthesia. Further, the model does not allow for primary tumor effects on metastases or co-evolution of the primary tumor and adaptive immune response. Metastasis is a dynamic process and further longitudinal studies will be needed to fully understand the impacts of anesthetic agents beyond the discrete time points tested in this study. As another limitation, all mice were briefly anesthetized with low dose isoflurane during imaging. This could introduce additional effects, but such effects would occur in all mice regardless of original anesthesia treatment. Furthermore, while preclinical studies offer well controlled experimental designs, their data is limited in scope and not directly applicable to humans. The results found in NSG and C57BL/6 mice may not be directly applicable to other mouse strains. In this study, we did not examine circulating levels of immune cells in the blood or whole blood counts prior to and after anesthetic exposure to rule out other factors that could have affected immune cell migration such as homeostatic differences or variance in baseline levels of inflammation among the mice. Finally, immunohistochemistry staining provides data only on the presence of certain immune cells but does not provide information on their functionality.

The current study provides compelling evidence that the immune effects of isoflurane and ketamine may be more important than other known or hypothesized mechanisms of increased tumor growth potential in murine models, such as increased VEGF and HIFs in the case of volatile anesthetics. Furthermore, in our study the overall quantity of metastases in the immunocompromised mice was significantly higher by several orders of magnitude at all time points compared with the metastatic burden seen in the isoflurane and ketamine immunocompetent groups. This suggests that immunoediting as it relates to tumor elimination is impaired but not eliminated by these agents.

Although our results suggest a temporary impairment of immune function, any potential clinical relevance must await prospective randomized clinical trials that assess immune function together with examination of the potential for anesthetic choice to influence cancer recurrence and metastasis following "major" surgeries [21].

Future experiments could investigate the temporal aspects of pro-metastatic effects by administering anesthetics at differing timepoints before or after tail vein injection to evaluate the response. Functional immunoassays could provide additional understanding for how different anesthetic agents alter immune cell function. For example, we could examine downstream signaling pathways such as NMDA receptor blockade by ketamine or alpha-2 blockade by xylazine with qPCR from CD3+ T cells and CD11b+ macrophages. Use of more complex mouse tumor models such as the 4T1 breast model of primary tumor and metastasis [40] may help to better understand anesthetic effects on the entirety of the metastatic cascade and incorporate primary tumor effects on adaptive immune function and metastases. These experiments may also include surgical intervention on the primary tumor in mice to more closely simulate procedures performed in human medicine.

## Supporting information

**S1 Data.**
(CSV)

**S2 Data.**
(CSV)

## Author Contributions

**Conceptualization:** Dominique Abrahams, Arig Ibrahim-Hashim, Robert A. Gatenby, Aaron R. Muncey.

**Data curation:** Dominique Abrahams, Arig Ibrahim-Hashim.

**Formal analysis:** Dominique Abrahams, Arig Ibrahim-Hashim, Robert S. Ackerman, Joel S. Brown, Christopher J. Whelan, Megan B. Garfinkel, Robert A. Gatenby, Aaron R. Muncey.

**Investigation:** Dominique Abrahams, Arig Ibrahim-Hashim, Aaron R. Muncey.

**Methodology:** Dominique Abrahams, Arig Ibrahim-Hashim, Joel S. Brown, Christopher J. Whelan, Robert A. Gatenby, Aaron R. Muncey.

**Project administration:** Arig Ibrahim-Hashim, Aaron R. Muncey.

**Software:** Christopher J. Whelan.

**Supervision:** Arig Ibrahim-Hashim.

**Validation:** Arig Ibrahim-Hashim.

**Writing – original draft:** Dominique Abrahams, Arig Ibrahim-Hashim, Robert S. Ackerman, Joel S. Brown, Christopher J. Whelan, Robert A. Gatenby, Aaron R. Muncey.

**Writing – review & editing:** Dominique Abrahams, Arig Ibrahim-Hashim, Robert S. Ackerman, Joel S. Brown, Christopher J. Whelan, Megan B. Garfinkel, Robert A. Gatenby, Aaron R. Muncey.

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
