## [Decision Letter · Decision Letter 0]

2 Aug 2023

PONE-D-23-19934Immunomodulatory and Pro-Oncologic Effects of Ketamine and Isoflurane Anesthetics in a Murine ModelPLOS ONE

Dear Dr. Muncey,

Thank you for submitting your manuscript to PLOS ONE. After careful consideration, we feel that it has merit but does not fully meet PLOS ONE’s publication criteria as it currently stands. Therefore, we invite you to submit a revised version of the manuscript that addresses the points raised during the review process.

The manuscript "Immunomodulatory and Pro-Oncologic Effects of Ketamine and Isoflurane Anesthetics in a Murine Model" presents well-written finding investigating the effects of isoflurane and ketamine on lung tumor growth in mice. The results demonstrate significant differences in metastases between immunodeficient and immunocompetent mice, with metastases being highest in the ketamine group among immunocompetent mice.  The authors should address limitations, consider potential bias in statistical analyses, provide evidence for decreased immune cell infiltration in blood, and explore downstream signaling pathways in CD3-positive T-cells and CD11b macrophages post-anesthesia injection to support their findings.

We look forward to receiving your revised manuscript.

Kind regards,

Syed M. Faisal, Ph.D.

Academic Editor

PLOS ONE

Journal Requirements:

3. We notice that your supplementary [figures/tables] are included in the manuscript file. Please remove them and upload them with the file type 'Supporting Information'. Please ensure that each Supporting Information file has a legend listed in the manuscript after the references list.

Reviewers' comments:

Reviewer's Responses to Questions

**Comments to the Author**

1. Is the manuscript technically sound, and do the data support the conclusions?

Reviewer #1: Yes

Reviewer #2: Yes

2. Has the statistical analysis been performed appropriately and rigorously? 

Reviewer #1: No

Reviewer #2: No

3. Have the authors made all data underlying the findings in their manuscript fully available?

Reviewer #1: Yes

Reviewer #2: No

4. Is the manuscript presented in an intelligible fashion and written in standard English?

Reviewer #1: Yes

Reviewer #2: Yes

5. Review Comments to the Author

Reviewer #1: The manuscript by Dominique Abrahams et al., titled "Immunomodulatory and Pro-Oncologic Effects of Ketamine and Isoflurane Anesthetics in a Murine Model," has been well written and study aimed to investigate the effects of isoflurane and ketamine, two commonly used anesthetics, on lung tumor growth in a murine model. The researchers hypothesized that these anesthetics would promote tumor growth by altering immune function. Lewis lung carcinoma cells were injected into immunocompetent and immunodeficient mice during exposure to anesthetics or without anesthesia. The mice were imaged over a 14-day period, and tumor growth and metastasis were quantified. The results showed that metastases were significantly higher in immunodeficient mice compared to immunocompetent mice. However, among immunocompetent mice, metastases were greatest in the ketamine group, followed by the isoflurane group, and least in the control group. Additionally, author showed immunocompetent mice receiving anesthetics also exhibited decreased infiltration of T lymphocytes and monocytes compared to control mice.

Limitations of this study: The given information mentions two mouse strains, C57BL/6 and NSG, as subjects for the experiment. It is important to consider that the results and conclusions drawn from these specific strains may not be directly applicable to other mouse strains or human subjects. Additionally, Metastasis is a dynamic process, and a single time point may not capture the full metastatic progression. Hence, time points and longitudinal studies are essential to understand the temporal dynamics of metastasis and the potential impact of anesthesia.

Major comments:

1) The statistical analysis results presented in Fig. 2 and Fig. 3 do not contain any detail on the number of technical or biological replicates used in the bar diagrams, including F-values and p-values. However, the specific details of the statistical analyses and the potential adjustments for multiple comparisons or statistical assumptions are not provided. It is important to ensure appropriate statistical methods and to consider potential sources of bias or error in the statistical analysis.

2) The author used IHC to demonstrate a decrease in immune cell infiltration in the lungs of WT/NSG mice but failed to provide any evidence that the overall number of immune cells in the mice's blood before or after anesthesia was also decreasing or remained unchanged, which may be a contributing factor for low immune cell migration. What are the other factors that may have contributed to the decreased migration in the lungs need to be discussed?

3) Please check overall blood count before and after anesthesia in WT/NSG mice.

4) Ketamine is a dissociative anesthetic that works by blocking the NMDA receptor, while xylazine is a sedative that works by binding to alpha-2 adrenoceptors. The author can check the effect on these downstream signaling pathways with qPCR from CD3-positive T-cells and CD11b macrophages for its direct activation post-anesthesia injection at day 0 and day14, to show a direct effect on immune cells.

Reviewer #2: This comparative study has shown clear cut effects of volatile and intravenous anaesthesia on the metastases of immunocompromised NSG and immunocompetent C57BL/6 mice. The results meet the objectives of study as well. Therefore I recommend this manuscript for acceptance after revision in the light of comments given below.

1-LWhat is the promoter/reporter gene insert in LL/2 Luc M-98 cells for bioluminescent imaging?

2- Authors are suggested to. Explain Lewis lung carcinoma cells type (LL/2 Luc M-98) at its first appearance in the MS and then maintain the uniformity in language throughout the MS, like CTR or Control.

3- Mention the age and sex of experimental animals in MS.

4- I did not find any Supplemental Tables in pdf of MS I received for review.

5- There are no legends for Supplemental Figures in the MS. Hence it is difficult to understand the different image panels of lungs and kidneys of NGS and corresponding bars of increased scorings of CD11b and CD3.

6-What is F2,6 in line 357, page 23. Pl correct this.

7- Flux rate indicate the rate of metabolism and cell growth in tumours, however I found the approx. same trend of flux increase NGS control too (Fig 2B), whereas, there is clear cut difference in tumor growth in control and treatment groups (Fig 1). Explain briefly in discussion.

6. PLOS authors have the option to publish the peer review history of their article (what does this mean?). If published, this will include your full peer review and any attached files.

Reviewer #1: **Yes: **ZEESHAN AHMAD

Reviewer #2: **Yes: **Khursheed Ali

---

## [Author Response · Author response to Decision Letter 0]

18 Sep 2023

Reviewer #1: The manuscript by Dominique Abrahams et al., titled "Immunomodulatory and Pro-Oncologic Effects of Ketamine and Isoflurane Anesthetics in a Murine Model," has been well written and study aimed to investigate the effects of isoflurane and ketamine, two commonly used anesthetics, on lung tumor growth in a murine model. The researchers hypothesized that these anesthetics would promote tumor growth by altering immune function. Lewis lung carcinoma cells were injected into immunocompetent and immunodeficient mice during exposure to anesthetics or without anesthesia. The mice were imaged over a 14-day period, and tumor growth and metastasis were quantified. The results showed that metastases were significantly higher in immunodeficient mice compared to immunocompetent mice. However, among immunocompetent mice, metastases were greatest in the ketamine group, followed by the isoflurane group, and least in the control group. Additionally, author showed immunocompetent mice receiving anesthetics also exhibited decreased infiltration of T lymphocytes and monocytes compared to control mice.

Limitations of this study: The given information mentions two mouse strains, C57BL/6 and NSG, as subjects for the experiment. It is important to consider that the results and conclusions drawn from these specific strains may not be directly applicable to other mouse strains or human subjects. Additionally, Metastasis is a dynamic process, and a single time point may not capture the full metastatic progression. Hence, time points and longitudinal studies are essential to understand the temporal dynamics of metastasis and the potential impact of anesthesia.

Thank you for noting these important limitations of our study, which we agree with. We have added prose indicating this on page 22 lines 483-484 and page 21, lines 477-479. We are currently in the planning stages of additional experiments focused on the temporal dynamics of metastasis in relation to anesthetic exposure. 

Major comments:

1) The statistical analysis results presented in Fig. 2 and Fig. 3 do not contain any detail on the number of technical or biological replicates used in the bar diagrams, including F-values and p-values. However, the specific details of the statistical analyses and the potential adjustments for multiple comparisons or statistical assumptions are not provided. It is important to ensure appropriate statistical methods and to consider potential sources of bias or error in the statistical analysis.

Thank you for pointing this out. We have updated the figure legends to include the information you have requested. We also changed the figures to a box-and-whisker format to allow the display of more statistical information. Furthermore, due to their importance we have moved the details of our statistical analysis (including tables) out of the supplemental content and into the main manuscript. We hope that this helps to clarify the logic and thought process of our statistical analysis.

2) The author used IHC to demonstrate a decrease in immune cell infiltration in the lungs of WT/NSG mice but failed to provide any evidence that the overall number of immune cells in the mice's blood before or after anesthesia was also decreasing or remained unchanged, which may be a contributing factor for low immune cell migration. What are the other factors that may have contributed to the decreased migration in the lungs need to be discussed?

This is an interesting and thoughtful comment. We have included this as a limitation of our study on page 22, lines 484-488 of the manuscript.

3) Please check overall blood count before and after anesthesia in WT/NSG mice.

While we are unable to go back and retroactively check these blood counts, we greatly appreciate this feedback and plan to use it in future work. For the current manuscript, as mentioned above, this will be listed as a limitation of our study.

4) Ketamine is a dissociative anesthetic that works by blocking the NMDA receptor, while xylazine is a sedative that works by binding to alpha-2 adrenoceptors. The author can check the effect on these downstream signaling pathways with qPCR from CD3-positive T-cells and CD11b macrophages for its direct activation post-anesthesia injection at day 0 and day14, to show a direct effect on immune cells.

Thank you for this great suggestion. We appreciate the feedback and will plan to do this in future experiments. We mentioned that a limitation of our study was the lack of functional immune cell testing on page 22, lines 489-490. And we have added prose mentioning plans to incorporate your suggestion into future work on page 23, lines 510-513

Reviewer #2: This comparative study has shown clear cut effects of volatile and intravenous anaesthesia on the metastases of immunocompromised NSG and immunocompetent C57BL/6 mice. The results meet the objectives of study as well. Therefore I recommend this manuscript for acceptance after revision in the light of comments given below.

1-LWhat is the promoter/reporter gene insert in LL/2 Luc M-98 cells for bioluminescent imaging?

Thanks for noting the absence of this information. The promotor is CMV and reporter is luciferase and this has been added on page 6, line 148. 

2- Authors are suggested to. Explain Lewis lung carcinoma cells type (LL/2 Luc M-98) at its first appearance in the MS and then maintain the uniformity in language throughout the MS, like CTR or Control.

Thank you for noticing this inconsistency, which has since been corrected.

3- Mention the age and sex of experimental animals in MS.

We have added this information on page 6, line 144.

4- I did not find any Supplemental Tables in pdf of MS I received for review.

Thanks for noting this. I am not sure what happened to that material in the submission. But we have now made the decision to include everything in the main manuscript file including the tables. 

5- There are no legends for Supplemental Figures in the MS. Hence it is difficult to understand the different image panels of lungs and kidneys of NGS and corresponding bars of increased scorings of CD11b and CD3.

This should also be fixed now since the supplemental material has been merged into the main manuscript as mentioned above.

6-What is F2,6 in line 357, page 23. Pl correct this.

This is the value of the F-test, which was part of our statistical analysis using ANOVA. We notice now that we had inconsistency in reporting this and other statistical measures within the manuscript. We have uniformly removed this and other statistics from the prose, opting instead to include all the statistical reporting within the figure legends. We can certainly add the statistics back throughout the prose in the results section if desired by the editorial office.

7- Flux rate indicate the rate of metabolism and cell growth in tumours, however I found the approx. same trend of flux increase NGS control too (Fig 2B), whereas, there is clear cut difference in tumor growth in control and treatment groups (Fig 1). Explain briefly in discussion.

Thanks for this astute observation. It made us realize that we had formatted figure 2B (now figure 3B since revision) in a suboptimal manner. We believe it is more visually intuitive now, with time in days on the bottom and bar colors corresponding to the different anesthetic treatments (opposite of how it was before). This way the reader’s eye does not have to jump between each trio of bars to see the anesthetic effect. Additionally, we changed the format for Figures 2 & 3 (now Figures 3 & 4 since revision) to a box and whisker plot, which conveys more information. We also added text on page 15, lines 343-344 clarifying that there was no statistically significant difference in flux amongst the different anesthetic treatments at each of the three time points in the NSG mice. One additional point to note is that due to the very large overall difference in flux between NSG and C57BL/6 mice, the only way to get all the data on one scale for comparison was to use a logarithmic scale. This compresses the effect visually, which makes it less pronounced, but we do not know of a better way to include all the data on one figure.

---

## [Editor Report · Decision Letter 1]

21 Sep 2023

Immunomodulatory and Pro-Oncologic Effects of Ketamine and Isoflurane Anesthetics in a Murine Model

PONE-D-23-19934R1

Dear Dr. Muncey,

We’re pleased to inform you that your manuscript has been judged scientifically suitable for publication and will be formally accepted for publication once it meets all outstanding technical requirements.

Kind regards,

Syed M. Faisal, Ph.D.

Academic Editor

PLOS ONE
---

## [Editor Report · Acceptance letter]

2 Oct 2023

PONE-D-23-19934R1 

Immunomodulatory and Pro-Oncologic Effects of Ketamine and Isoflurane Anesthetics in a Murine Model 

Dear Dr. Muncey:

I'm pleased to inform you that your manuscript has been deemed suitable for publication in PLOS ONE. Congratulations! Your manuscript is now with our production department. 

Kind regards, 

on behalf of

Dr. Syed M. Faisal 

Academic Editor

PLOS ONE